# Microstructural Modeling of Rheological Mechanical Response for Asphalt Mixture Using an Image-Based Finite Element Approach

**DOI:** 10.3390/ma12132041

**Published:** 2019-06-26

**Authors:** Wenke Huang, Hao Wang, Yingmei Yin, Xiaoning Zhang, Jie Yuan

**Affiliations:** 1School of Civil Engineering, Guangzhou University, Guangzhou 510006, China; 2Department of Civil and Environmental Engineering, Rutgers, The State University of New Jersey, New Brunswick, NJ 08854, USA; 3School of Civil and Transportation Engineering, Guangdong University of Technology, Guangzhou 510006, China; 4School of Civil Engineering and Transportation, South China University of Technology, Guangzhou 510641, China

**Keywords:** asphalt mixture, rheological, viscoelastic, microstructure, numerical simulations

## Abstract

In this paper, an image-based micromechanical model for an asphalt mixture’s rheological mechanical response is introduced. Detailed information on finite element (FE) modeling based on X-ray computed tomography (X-ray CT) is presented. An improved morphological multiscale algorithm was developed to segment the adhesive coarse aggregate images. A classification method to recognize the different classifications of the elemental area for a confining pressure purpose is proposed in this study. Then, the numerical viscoelastic constitutive formulation of asphalt mortar in an FE code was implemented using the simulation software ABAQUS user material subroutine (UMAT). To avoid complex experiments in determining the time-dependent Poisson’s ratio directly, numerous attempts were made to indirectly obtain all material properties in the viscoelastic constitutive model. Finally, the image-based FE model incorporated with the viscoelastic asphalt mortar phase and elastic aggregates was used for triaxial compressive test simulations, and a triaxial creep experiment under different working conditions was conducted to identify and validate the proposed finite element approach. The numerical simulation and experimental results indicate that the three-dimensional microstructural numerical model established can effectively analyze the material’s rheological mechanical response under the effect of triaxial load within the linear viscoelastic range.

## 1. Introduction

Asphalt mixture is the most widely used road construction material in the paving industry. The mechanical behavior of the asphalt mixture is complex due to the heterogeneity of asphalt’s composite materials and the time- and temperature-dependent viscoelastic binder. In the aim of optimizing materials design, it is necessary to study the behavior of this composite material.

Asphalt mixture can be considered to consist of three phases: the mortar phase (asphalt cement with a filler that is smaller than 2.36 mm in size), the aggregate phase, and the air voids [1,2,3]. Thus, some researchers have proposed analytically-based models to investigate the rheological response for asphalt mixture from a micromechanical perspective [4,5,6]. The non-destructive X-ray computed tomography (CT) technique is used as an effective method to capture images of the internal microstructure of asphalt mixture [7,8,9,10,11,12,13]. There have been a number of recent attempts to use X-ray CT images to reconstruct a three-dimensional finite element specimen to predict the mechanical properties of asphalt mixture [14,15,16,17,18,19]. The Finite Element Analysis (FEA) with an X-ray CT method may be able to more accurately predict the behavior and failure mechanisms in asphalt mixture with regards to classical analytical solutions. The image-based finite element method requires image analysis techniques in order to segment the CT slices consisting of different compositions, the mechanical properties of different phases, and the use of mechanical testing to identify the material’s parameters. The majority of these microstructure numerical samples for asphalt mixture face challenges with regard to the accurate geometry of different phases and the precise description of asphalt mortar behavior. Asphalt pavement is in a complicated and three-dimensional stress state when subjected to traffic loading. The triaxial test is an effective way to obtain the mechanical response of asphalt pavement because the stress conditions in the triaxial test conform to those of the asphalt pavement [20].

However, the majority of previous studies were mainly limited to uniaxial compression tests, indirect tensile tests (IDTs), or bending beam tests [21,22,23]. The absence of complex loading conditions makes them less able to reflect the real mechanical response of asphalt mixture. In order to establish a three-dimensional finite element approach that reflects the real internal microstructure of asphalt mixture and simulates the actual response of asphalt mixture under traffic loading, this paper presents an image-based microstructure modeling approach for simulating the triaxial compressive mechanical response of asphalt mixture at a given temperature. Firstly, a voxel-based three-dimensional digital reconstruction model of the asphalt concrete specimen is constructed based on the X-ray images. Then, the three-dimensional viscoelastic constitutive model is applied to better describe the mechanical behavior of asphalt mortar and is implemented in a finite element code using the ABAQUS user material subroutine (UMAT). After that, the image-based numerical specimen combining with the ABAQUS user material subroutine is applied to predict the triaxial compressive mechanical behavior of a real microstructure asphalt mixture specimen. Finally, a triaxial compressive experiment was conducted to identify and validate the proposed finite element approach compared with the predicted results.

## 2. Finite Element Method Based on a CT Image 

Asphalt mixture is a particulate composite material consisting of aggregate, mortar, and air voids. In this section, a cylindrical Hot Mixture Asphalt (HMA) mixture (AC-20) lab specimen was prepared for capturing the internal microstructure with a non-destructive industrial X-ray CT technique. Then, the grayscale thresholds for dividing aggregate, mortar, and air voids were chosen based on the improved OTSU image processing method. Additionally, a voxel-based three-dimensional image model of the specimen was constructed. In Section 2.1 and Section 2.2, the reconstructed process of the microstructure model and the numerical implementation of the mortar’s constitutive model are described in detail.

### 2.1. Microstructural FE Model Construction

#### 2.1.1. Microstructure Acquisitions

X-ray CT is an effective method to capture the internal microstructure of asphalt mixture. Industrial CT systems consist of the following basic components: a radiation source, radiation detector, collimator, data-collecting system, mechanical specimen-scanning system, computer system (hardware and software), and auxiliary system (auxiliary power supply and auxiliary security system). In this study, the YXLON Compact-225 X-ray scanner (YXLON International, Hamburg, Germany) is used to obtain a detailed microscopic structure of the asphalt mixture specimen. The highest spatial resolution of the YXLON Compact-225 X-ray scanner can reach 20 microns. 

Scanned CT images of asphalt mixture have different grayscale intensities between 0 and 255, where denser materials have a higher intensity. The pixel grayscale intensities of the aggregate are close to 255. The pixel grayscale intensities of air voids and the background are closed to 0. The pixel grayscale intensities of the mortar are between those of the aggregates and air voids. The OTSU method was utilized for the segmentation of the AC-20 microstructure.

Particles that are connected each other are a common phenomenon in the segmentation process of X-ray CT slices. These particles increase the difficulty of numerical modeling [24,25], because the finite element code will treat the connected particles as isolated ones. To overcome this difficulty, we developed an improved morphological multiscale algorithm based on several different sizes of structural elements to segment coarse aggregate adhesion images. The algorithm flow chart, which was implemented in the MATLAB, is shown in Figure 1.

The flow of the algorithm is briefly described as follows: Circular shaped structure elements with a radius of 1, 2, 3, and 4 were selected to separate every likely connected particle based on the morphological multiscale algorithm. Each likely connected image operated four times, and for those four kinds of segmentation, divided line numbers during every morphological multiscale operation process were recorded. Segmentation of the maximum frequency of the divided line numbers that appeared in the four times operation was considered practical segmentation of this binary image and segmentation with a minimal structural element corresponding to the segmented image was treated as the practical image segmentation. Every marker, including the isolated particle image and the likely connected image after the morphological multiscale operation, was merged into one image for reconstruction.

#### 2.1.2. Digital Samples Generation

Every binary image is represented by an m × n logical mortar where pixel values are 1 (true) or 0 (false). A voxel-based three-dimensional digital reconstruction model of the asphalt mixture specimen is constructed. In order to input the element and node information into the finite element software, such as ABAQUS, the node and element numbering rules for generating the pixel-based numerical model are defined as follows. The node and element numbering rules are: Nodes and element numbering start from the lower left corner of the mortar, and then go to the right side of each line; the last number of each line is followed by the next line. The position of the corresponding element number in the first image differs from that of the adjacent image by m × n. While the position of the corresponding node number in the first image differs from that of the adjacent image by (m + 1)(n + 1). The element and node information is generated with the MATLAB programming and written into the input file of ABAQUS for numerical simulation.

### 2.2. Asphalt Mortar Constitutive Model

In this section, the linear viscoelastic constitutive model used to represent the behavior of the asphalt mortar phase is introduced. Then, the incremental formulations of the constitutive model implemented in finite element code are developed in detail.

#### 2.2.1. Linear Viscoelastic Constitutive Model

For linear viscoelastic materials, including asphalt mortar, the stress–strain constitutive relation is expressed by convolution integrals. The general Maxwell model is usually utilized for studying the viscoelastic behaviors of the asphalt matrix. The general viscoelastic model consists of several Maxwell elements in parallel, as shown in Figure 2.

In the case of a stress response at constant strain, the convolution relations is explained as follows for one-dimensional problems:(1)$$\sigma \left(t\right)=E_{\infty }\varepsilon +\int_{0}^{t}\Delta E\left(\varphi^{t}-\varphi^{\tau }\right)\frac{d\left(\varepsilon \right)}{d\tau }d\tau$$
where the $E_{\infty }$ is the instantaneous elastic modulus, *t* is current time, and *τ* is integration time variable; $\varphi^{t}$ is the reduced time at current time, $\varphi^{\tau }$ is the reduced time at integration time, and ΔE is the transient modulus. This is given by:(2)$$\Delta E\left(\varphi \right)=\overset{N}{\underset{n=1}{\sum }}E_{n}\exp\left(-\rho_{n}\varphi \right)$$
where $E_{n}$ is the nth coefficient of the Prony series and $\rho_{n}$ is the nth relaxation time.

For two-dimensional and three-dimensional problems, Equation (1) can be decomposed into deviatoric and volumetric components, such that:(3)$$S_{ij}^{t}=2G_{0}e_{ij}^{t}+2\int_{0}^{t}\Delta G\left(\varphi^{t}-\varphi^{\tau }\right)\frac{d\left(e_{ij}^{\tau }\right)}{d\tau }d\tau$$
(4)$$\sigma_{kk}^{t}=3K_{0}\varepsilon_{kk}^{t}+3\int_{0}^{t}\Delta K\left(\varphi^{t}-\varphi^{\tau }\right)\frac{d\left(\varepsilon_{kk}^{\tau }\right)}{d\tau }d\tau$$
(5)$$\sigma_{ij}^{t}=S_{ij}^{t}+\frac{1}{3}\sigma_{kk}^{t}\delta_{ij}$$
where $S_{ij}^{t}$ and $\sigma_{kk}^{t}$ are the deviatoric stress tensor and volumetric stress tensor, respectively; $e_{ij}^{t}$ and $\varepsilon_{kk}^{t}$ are the are the deviatoric strain and the volumetric strain, respectively; $\sigma_{ij}^{t}$, $G_{0}$ and $K_{0}$ are the total stress tensor, the instantaneous effective elastic shear modulus and bulk modulus, respectively; ΔG and ΔK are the transient shear modulus and bulk modulus, respectively.

In most cases, the asphalt mortar Poison’s ratio *ν* is assumed to be time-independent, allowing for all the material properties in Equations (3)–(5) to be determined from the data of uniaxial tensile tests as follows:(6)$$J_{0}=2\left(1+\nu \right)D_{0}$$
(7)$$B_{0}=3\left(1-2\nu \right)D_{0}$$
(8)ΔJ(φ)=2(1+ν)ΔD(φ)
(9)ΔB(φ)=3(1−2ν)ΔD(φ)
where $D_{0}$ and ΔD(φ) are the instantaneous effective elastic tensile compliance and the transient tensile compliance, respectively. $J_{0}$ and $B_{0}$ are the instantaneous effective elastic shear and bulk compliances, respectively; ΔJ(φ) and ΔB(φ) are the transient shear compliance and bulk compliance, respectively.

In most cases, the asphalt mortar Poison’s ratio *ν* is assumed to be time-independent, such that all the material properties in Equations (3)–(5) can be determined from the data of uniaxial tensile tests. However, for asphalt mortar, the Poisson’s ratio ν is not a material constant but can depend upon time, temperature, and loading frequency. Although much progress has been made in experimental tests and numerous studies for determining the time-dependent Poisson’s ratio of linear viscoelasticity, this field is still fraught with considerable difficulty. To avoid complicated experiments in determining the time-dependent Poisson’s ratio directly, we have made numerous attempts to obtain all the material properties in Equations (3)–(5) indirectly and validated the parameters [26]. According to the elastic–viscoelastic correspondence principle, each of the tensile creep compliance *D*, the shear compliance *J* and bulk compliance *B* can be obtained from the other two using Laplace transform. Thus, the tensile creep compliance D, the shear compliance J can be easily determined directly from uniaxial tensile tests and torsion tests, respectively. The fundamental compliances of *D* and *J* are given by:(10)$$B_{0}=9D_{0}-3J_{0}$$
(11)ΔB(φ)=9ΔD(φ)−3ΔJ(φ)
(12)$$\Delta J\left(\varphi \right)=\overset{N}{\underset{n=1}{\sum }}J_{n}\left(1-\exp\left(-\lambda_{n}\varphi \right)\right)$$
where $D_{n}$ is the nth coefficient of the Prony series and $\lambda_{n}$ is the nth retardation time.

#### 2.2.2. Numerical Implementation for Asphalt Mortar

The finite element method (FEM) is actually an incremental approach for numerical analysis. Current stress and strain at the integration points of each element at every time increment are obtained from the stress and strain over the previous loading history. Thus, the incremental deviatoric and volumetric formulations can be derived with some algebraic manipulations.

According to Equations (3)–(5), the three-dimensional incremental deviatoric and volumetric stress formulations with constant strain rate can be expressed as:(13)$$\Delta S_{ij}^{t}=S_{ij}^{t}-S_{ij}^{t-\Delta t}=G\left(t\right)\Delta e_{ij}+2\overset{N}{\underset{n=1}{\sum }}G_{n}\left[exp\left(-\rho_{n}\Delta t\right)-1\right]q_{ij,n}^{t-\Delta t}$$
(14)$$G\left(t\right)=2\left\{G_{0}-\overset{N}{\underset{n=1}{\sum }}G_{n}\frac{1}{\rho_{n}\Delta t}\left(exp\left(-\rho_{n}\Delta t\right)-1\right)\right\}$$
(15)$$q_{ij,n}^{t}=exp\left(-\rho_{n}\Delta t\right)q_{ij,n}^{t-\Delta t}+\Delta e_{ij}\frac{1-exp\left(-\rho_{n}\Delta t\right)}{\rho_{n}\Delta t}$$
(16)$$\Delta \sigma_{kk}^{t}=\sigma_{kk}^{t}-\sigma_{kk}^{t-\Delta t}=K\left(t\right)\Delta \varepsilon_{kk}+3\overset{N}{\underset{n=1}{\sum }}K_{n}\left[exp\left(-\rho_{n}\Delta t\right)-1\right]q_{kk,n}^{t-\Delta t}$$
(17)$$K\left(t\right)=3\left\{K_{0}-\overset{N}{\underset{n=1}{\sum }}K_{n}\frac{1}{\rho_{n}\Delta t}\left(exp\left(-\rho_{n}\Delta t\right)-1\right)\right\}$$
(18)$$q_{kk,n}^{t}=exp\left(-\rho_{n}\Delta t\right)q_{kk,n}^{t-\Delta t}+\Delta \varepsilon_{kk}\frac{1-exp\left(-\rho_{n}\Delta t\right)}{\rho_{n}\Delta t}$$
(19)$$\Delta \sigma_{ij}^{t}=\Delta S_{ij}^{t}+\frac{1}{3}\Delta \sigma_{kk}^{t}\delta_{ij}$$
where $\Delta S_{ij}^{t}$ and $\Delta \sigma_{kk}^{t}$ are the incremental deviatoric and volumetric stress tensors at previous time *t*, respectively. $q_{ij,n}^{t}$ and $q_{kk,n}^{t}$ are the shear and volumetric hereditary integrals, respectively. The hereditary integrals are updated at the end of every converged time increment.

For the initial increment, the variables $q_{ij,1}^{t}$ and $q_{kk,1}^{t}$ are set to $\Delta S_{ij}\frac{1-exp\left(-\lambda_{n}\Delta t\right)}{\lambda_{n}\Delta t}$ and $\Delta \sigma_{kk}\frac{1-exp\left(-\lambda_{n}\Delta t\right)}{\lambda_{n}\Delta t}$, respectively. $\Delta \sigma_{ij}^{t}$ is the total incremental stress.

This three-dimensional numerical constitutive model is implemented within the FE code using FORTRAN. The ABAQUS user material subroutine (UMAT) is applied for this purpose. The flowchart for this procedure is illustrated in Figure 3.

## 3. Experimental Scheme

### 3.1. Asphalt Mortar Parameters Determination

Asphalt mortar is comprised of fine aggregates and asphalt binder. The aggregate gradation for asphalt mortar was extracted from the HMA mixture (AC-20) gradation without coarse aggregates (larger than 2.36 mm) as shown in Table 1. Asphalt mortar specimens were constructed using diabase aggregate and styrene-butadiene-styrene (SBS) modified asphalt binders. The asphalt binder content in asphalt mortar is the same as the AC-20, excluding the binder absorbed by coarse aggregates. The asphalt content used in asphalt mortar was 13.2%.

The prepared asphalt mortar specimen has a dimension of 10 mm × 10 mm × 50 mm in length, width and height, respectively. Uniaxial tensile tests and torsion tests were applied to determine all the material parameters in three-dimensional viscoelastic constitutive model at a temperature of 20 °C. The results of Prony series coefficients are shown in Table 2. After the parameters are determined, the viscoelastic numerical constitutive model can be used for numerical analysis.

### 3.2. Triaxial Creep Experiment

An AC-20-type asphalt mixture specimen was adopted in the experiment, of which the raw material and mineral aggregate gradation were consistent with that reported in Section 3.1. An HYD-25 servo hydraulic material testing machine (Cooper Research Technology Co Ltd., Derbyshire, UK) was used in the triaxial creep experiment as shown in Figure 4. The end faces of each specimen were coated with a thin lubricating membrane of polytetrafluoroethylene in order to eliminate the influence of end face friction.

The three-dimensional linear viscoelastic constitutive model of asphalt mortar indicates that the asphalt mixture specimen is not allowed to be destroyed during the test, and the applying load should be sufficiently small. Thus, mortar deformation is within the scope of linear viscoelasticity. Therefore, both confining pressure and axial load should be kept within a small range before the triaxial creep test. The compression strength failure test of the samples should be implemented first to confirm the loading scope. The loading scheme of the triaxial creep test in this study is shown in Table 3.

Considering the distribution property of the asphalt mixture specimen, and to avoid repeated experiments, the loading mode shown in Figure 5 was adopted by the triaxial creep experiment in this study. The loading mode is as follows. The first-order load is simultaneously implemented when confining pressure constancy occurs. Once the first-order load was finished, it was completely unloaded. Then, the second-order load was implemented and completely unloaded once the loading was finished, and so on.

## 4. Numerical Analysis and Model Validation

### 4.1. Analysis of Triaxial Creep Numerical Simulation Result

Following the AC-20 asphalt mixture cylinder mixture specimen obtained by coring in Section 3.2, before the triaxial test, we collected a piece of image every 0.1 mm along the specimen height direction of the asphalt specimen by industrial CT. The CT slice resolution was 1500 × 1458. A total of 1000 slices of the asphalt mixture specimen with sizes of 100 mm (diameter) × 100 mm (height) were obtained. To reduce the element amounts in the three-dimensional numerical model, the CT slices were first compressed before reconstruction. The three-dimensional numerical model had a total number of 647,315 elements and 672,356 nodes after compression. The eight-node three-dimensional solid integration elements (C3D8) (with an element thickness) were used in constructing the numerical model. The microstructural numerical model of the asphalt mixture after reconstruction is shown in Figure 6. A rigid body was used to simulate the loading plates at the top and bottom of the specimen. The mortar material property that adopted the three-dimensional viscoelastic constitutive model is shown in Table 2. The numerical viscoelastic constitutive model is implemented within the FE code using FORTRAN.

The deformation of a coarse aggregate under loading was very small and only needed to ensure a sufficient modulus. The modulus of elasticity and the Poisson’s ratio for the coarse aggregate were assumed to be 25 GPa and 0.25, respectively. In the three-dimensional finite element model, the air void was presumed to be the constituent of the model, which can be regarded as the elastic material that has a sufficiently small modulus. In this study, the modulus of elasticity and the Poisson’s ratio for the air void were assumed to be 0.3 MPa and 0.35, respectively.

Once the asphalt mixture numerical specimen was established, the implementation of the lateral confining pressure was the key step for the numerical analysis.

The lateral surface of the numerical model is an unsmooth surface, which is constituted by considerable elemental areas (Figure 6). Therefore, when the confining pressure was placed on the lateral model, manually selecting the specimen side was difficult because of the large amount of the elemental area. Thus, a classification method to recognize different classifications of the elemental area was proposed in this study.

The sequence definition of the node and surface order for the three-dimensional solid element by the ABAQUS is shown in Figure 7a. According to the establishment method of the previous numerical model, the confining pressure was actually imposed on the four directions of the element surfaces in the numerical model presented in Figure 7b.

An algorithm to search for various types of elemental areas, which are able to rapidly obtain each type of element serial number, is proposed in this study. The algorithm flow is shown as follows. First, each CT slice was treated with binarization. Second, a search from bottom to top and from left to right was performed on the element surface of S3, S4, S5, and S6. When searching in each column, once the pixel value was 1 for the first time, we stopped searching in the column and continued with the next column until searches through all columns were finished. Finally, elements corresponding to the four types of element surfaces were formed respectively.

This algorithm was applied to search for the four element surfaces of the asphalt mixture numerical model with the confining pressure, and the search result was edited to the INP folder. In the elemental area of the S6 type, for instance, part of the schematic search results is shown in Figure 8.

The three-dimensional numerical model established by the above method was used to carry out the simulation tests in the loading modes shown in Table 3. The incremental steps were set to be 600. The deformation contours of the triaxial compression axial creep under different loading conditions are shown in Figure 9.

The figures indicate that the axial deformation contour was unlike the homogeneous deformation contour, which was shown as a random status based on the distribution structure of different materials in the asphalt mixture specimen, rather than being a linear gradient from the upper part to the lower part. We constructed the tendency of the axial creep deformation with a time change under different loading conditions. A data point was collected at every 6 s, which is shown in Figure 10. The figure indicates that the larger the axial pressure, the larger the deformation of the compression creep, and the axial deformation with the confining pressure was smaller than the deformation without confining pressure. The main reason for this result is because in the triaxial test, imposing confining pressure to the specimen significantly improved axial compressive resistance. Then, the deformation of the specimen that was triaxially compressed was smaller than the specimen that was homotaxially compressed under the same loading condition.

The creep compliance curve under different loading conditions is shown in Figure 11. The figure indicates that for the non-confining pressure condition, the creep compliance curve obtained by the axial deformation calculation coincided. Therefore, a linear relationship occurred between the axis deformation of the three-dimensional numerical model and the load without confining pressure in this study. When there was confining pressure, certain differences occurred for the three creep compliance curves, and the smaller the principal stress difference, the larger the difference. The maximum change of the creep compliance at 600 s was 3.26%, which can be regarded as the load having a linear relationship with deformation under the three loading modes in this experiment.

### 4.2. Triaxial Creep Experiment Result and Comparison

To validate the effectiveness of the numerical model, the triaxial creep experiments under different loading conditions were conducted based on the same methods described in Section 3.2. The relationship between creep deformation and time in experiment and simulation is shown in Figure 12.

The figures indicate that the experimental curves were close to the numerical curves under lower principal stress differences. However, as the difference of principal stress increases, the difference between experimental and numerical results becomes larger. The difference between the numerical and experimental results was comparatively large, especially when the axial load reached 160 kPa. The principal causes for the above deviations are as follows. Firstly, due to the errors accumulated in obtaining the parameters of the three-dimensional constitutive model for asphalt mortar, the simulation results deviate from the experimental results. Secondly, the interface between the aggregate and the asphalt mortar was not considered in this study. Instead, the interface was assumed to be fully bound. That is, the nodes at the interface were shared by aggregate and asphalt mortar during modeling. Therefore, interface deformation between the aggregate and asphalt mortar in simulation is different from that in practice. Thirdly, the main reason that the difference between experimental and numerical results iss correspondingly larger when principal stress difference is larger is that the creep compliance curve of the asphalt mortar is measured within a linear viscoelastic range. The conclusion in Section 4.1 indicates that regardless of whether confining pressure existed, the developmental trend of the simulated creep compliance curve was consistent. Therefore, for a numerical simulation, the load and deformation are linearly related. The experimental creep curves shown in Figure 13 indicate that when the axial load was 160 kPa, the development tendency of the creep compliance curve was significantly different from the creep compliance curve when the load was 80 kPa and 120 kPa. Therefore, complying with the theory of linear viscoelasticity, the creep compliance curve was consistent within the linear viscoelasticity scope. However, as the load increased, once the load was out of the linear viscoelasticity scope that the material could bear, the development tendency of the creep compliance curve showed different development forms.

Figure 13 further provides a rule that the experimental and simulation creep deformation rates change with time under different loading conditions. This figure indicates that the deformation rates for experiment and simulation needed different times to achieve stable values. The simulation results quickly stabilized because the instantaneous load was imposed by the numerical simulation. After the load was imposed on the specimen, the load could attain the predetermined load value. By contrast, the load imposed by the experiment could not immediately achieve the required value, and usually 5 to 10 s were needed to achieve the predetermined load value. Overall, the deformation rate became steady after 50 s, and the creep deformation rate reached a stable value.

In summary, when the load was 80 kPa and 120 kPa, the numerical results were close to the experimental results. This is mainly because the load was comparatively small, and the asphalt mixture specimen was within the linear viscoelastic scope. However, when the load was out of the linear viscoelastic scope, differences occurred between the measured and numerical results. Therefore, the asphalt mortar viscoelastic constitutive model established in this study and the numerical calculation method, as well as the three-dimensional numerical model based on voxels, can precisely predict the stress–strain response in compound loading within the linear viscoelastic scope.

## 5. Conclusions

Based on X-ray CT imaging and the voxel method, this study established a three-dimensional microstructure numerical model for asphalt mixture. Triaxial creep simulations under different working conditions were conducted using the proposed numerical model for asphalt mixture. Compared with the numerical and the experimental results, conclusions are drawn as follows:(1)An algorithm based on the improved morphological multiscale method was proposed to effectively split the adhesion aggregate in X-ray CT images. The voxel-based three-dimensional microstructure numerical model of asphalt mixture was successfully established using the segmented CT images.(2)Following the simple algebraic relation between the bulk compliance with tensional creep compliance and torsion creep compliance, we obtained the creep parameters of a three-dimensional viscoelastic constitutive model for asphalt mortar by simple tensile and torsion creep experiments. This avoids complicated experiments that directly measure the Poisson’s ratio.(3)Triaxial creep numerical experiments under different loading conditions were conducted by using the established three-dimensional microstructural model. A comparison between the numerical and experimental results indicate that the experimental results could be better fitted with the numerical within the linear viscoelastic scope. However, a certain deviation was found between the numerical and simulation results when the load was out of the linear viscoelastic scope.(4)The three-dimensional microstructural model established by this study was able to effectively analyze the material mechanical response under triaxial load within the linear viscoelastic scope. This can provide an effective numerical method for the aided design of asphalt mixture.

## Figures and Tables

**Figure 1 materials-12-02041-f001:**
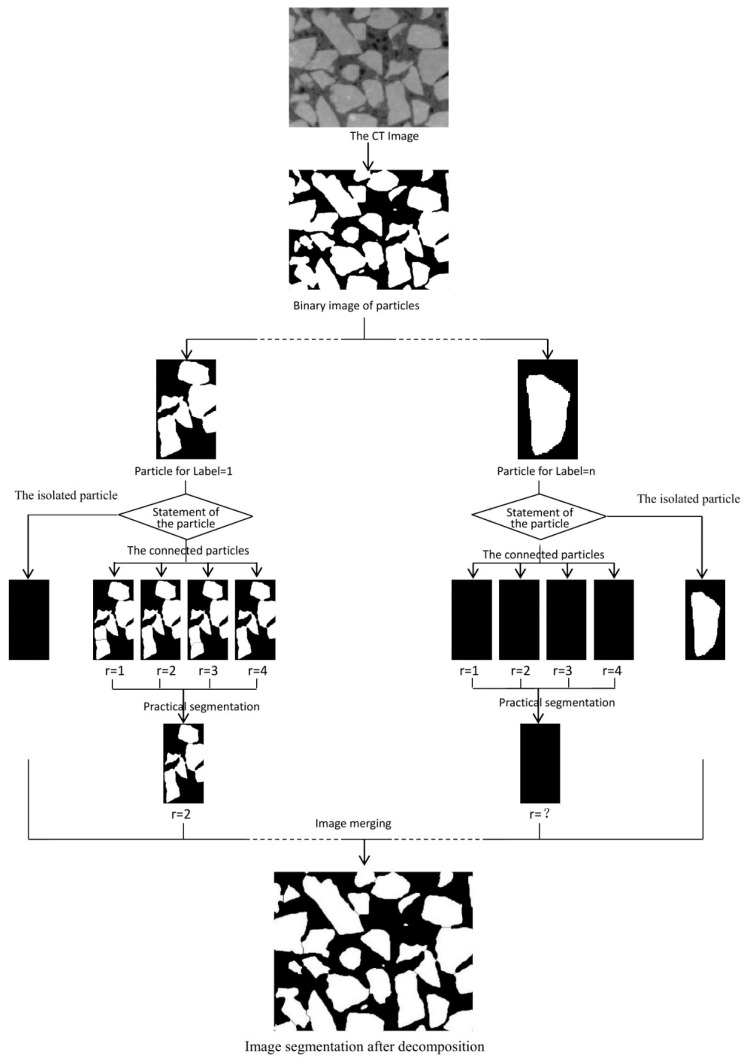
The morphological multiscale algorithm flow chart for adhesion aggregate segmentation.

**Figure 2 materials-12-02041-f002:**
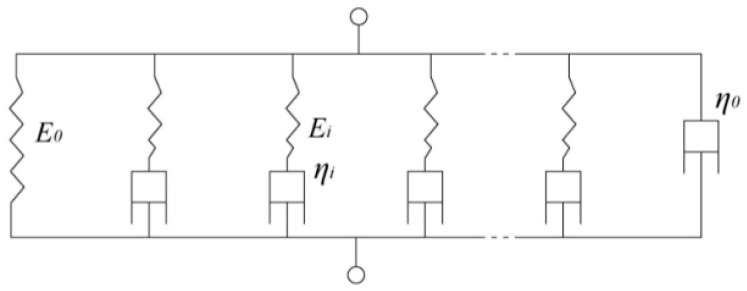
General Maxwell model for linear viscoelastic material.

**Figure 3 materials-12-02041-f003:**
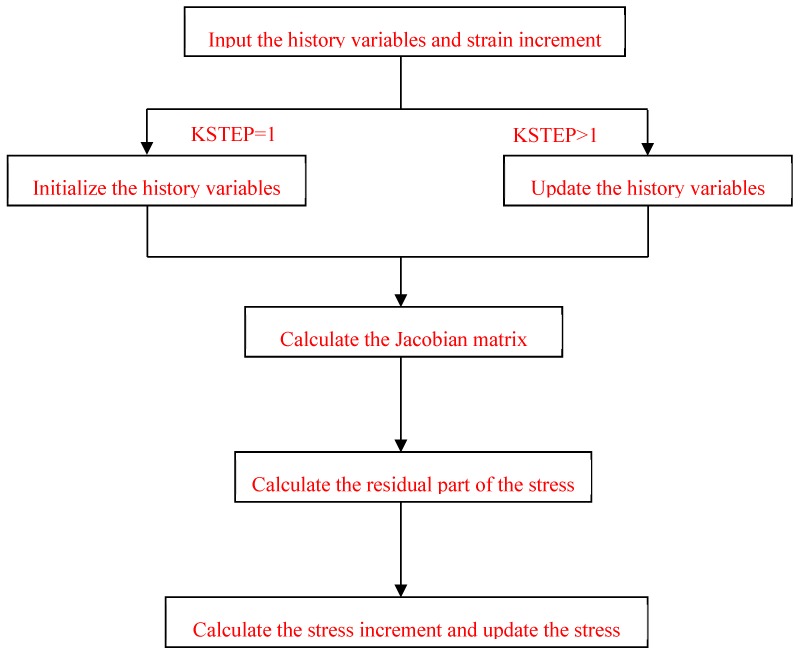
The flowchart for the three-dimensional numerical constitutive model derivation procedure.

**Figure 4 materials-12-02041-f004:**
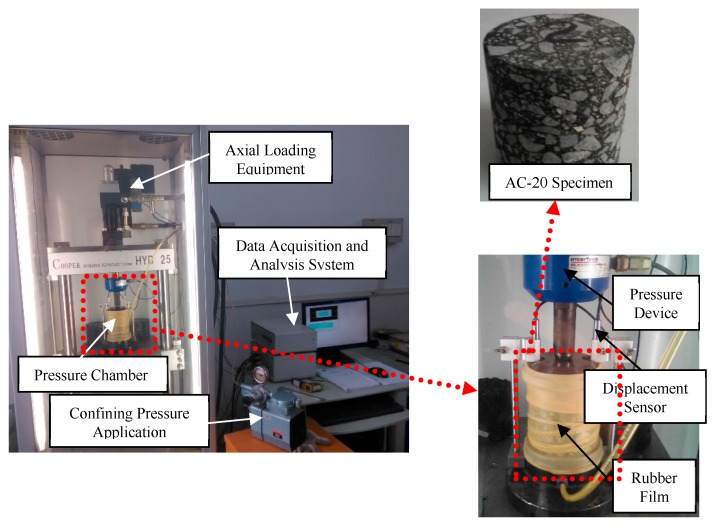
The basic composition of the HYD-25 triaxis instrument.

**Figure 5 materials-12-02041-f005:**
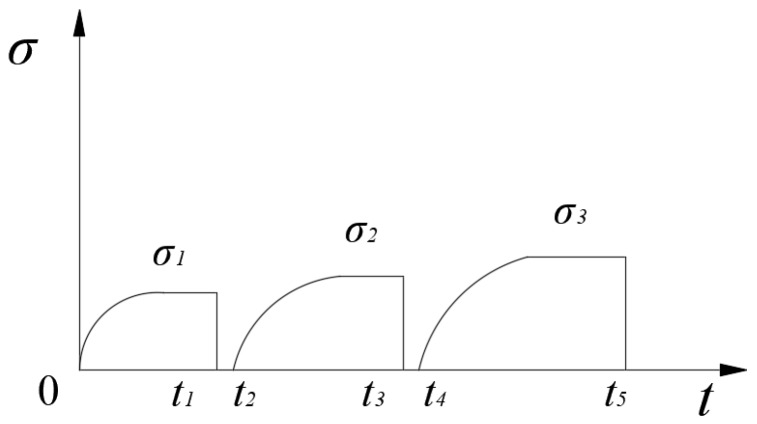
The proposed loading mode of the triaxial creep experiment.

**Figure 6 materials-12-02041-f006:**
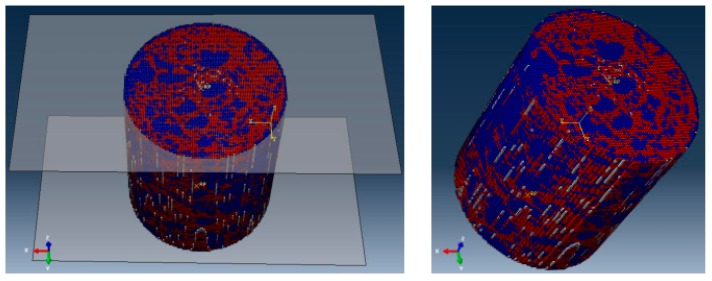
Three-dimensional finite element model of asphalt mixture.

**Figure 7 materials-12-02041-f007:**
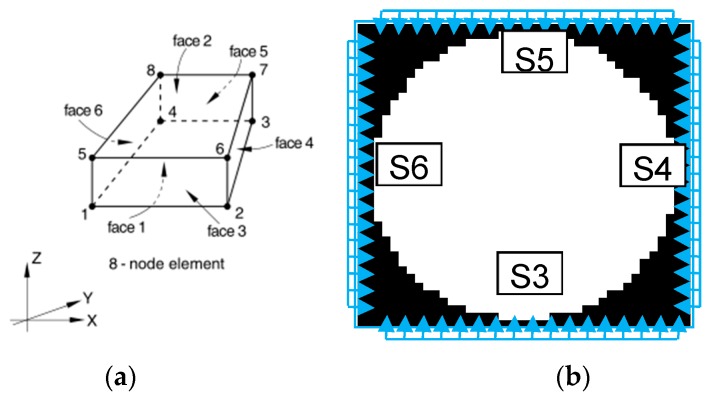
(**a**) Definition of node and surface order for three-dimensional solid element; (**b**) schematic diagram of the confining pressure of the cross section.

**Figure 8 materials-12-02041-f008:**
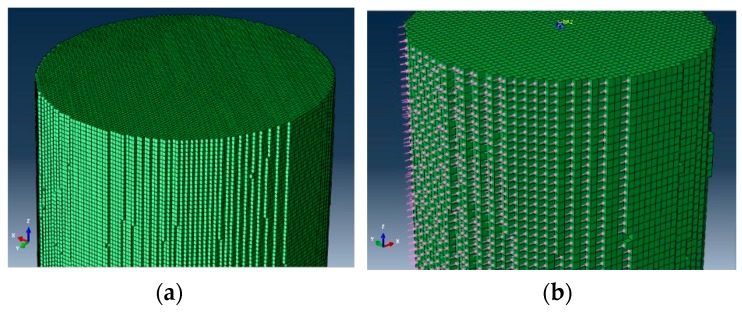
(**a**) Element side surface of the asphalt mixture numerical model; (**b**) S6 type element surface searching results.

**Figure 9 materials-12-02041-f009:**
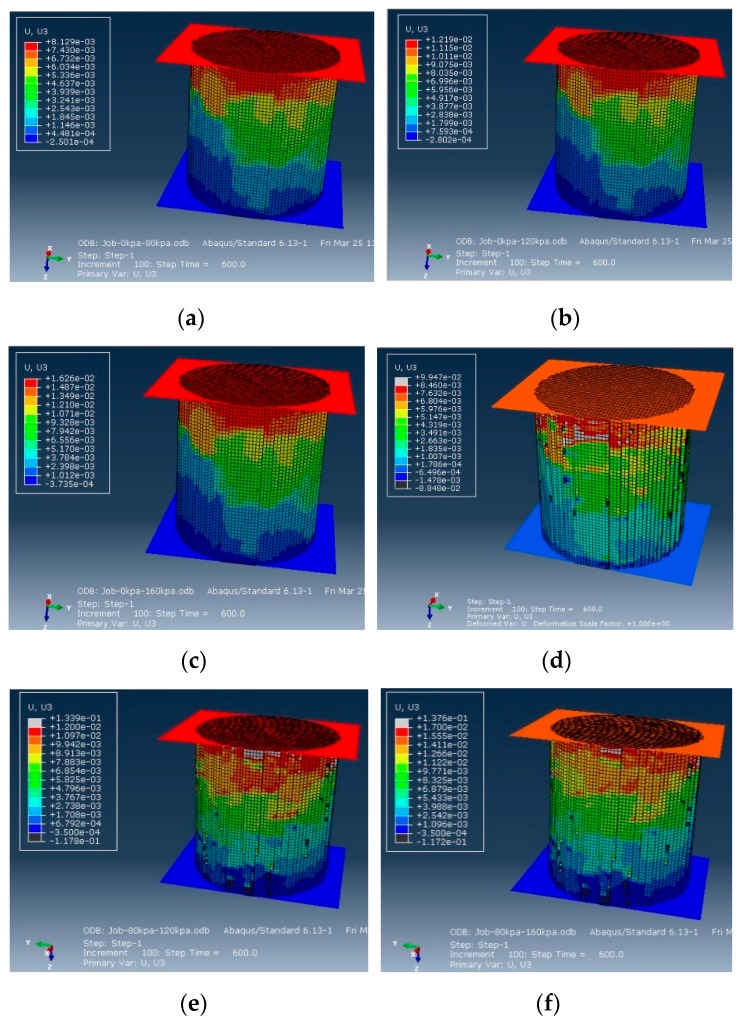
Creep deformation: (**a**) 80 kPa vertical load, without confining pressure; (**b**) 120 kPa vertical load, without confining pressure; (**c**) 160 kPa vertical load, without confining pressure; (**d**) 80 kPa vertical load, with 60 kPa confining pressure; (**e**) 120 kPa vertical load, with 60 kPa confining pressure; (**f**) 160 kPa vertical load, with 60 kPa confining pressure.

**Figure 10 materials-12-02041-f010:**
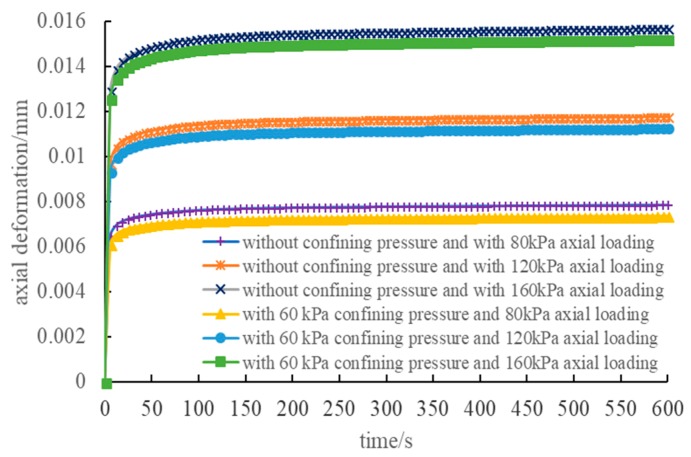
Axial deformation under different loading conditions.

**Figure 11 materials-12-02041-f011:**
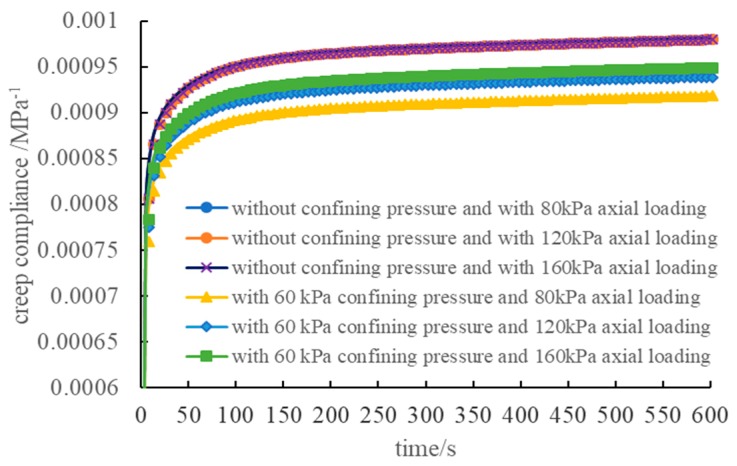
Creep compliance curves under different loading conditions.

**Figure 12 materials-12-02041-f012:**
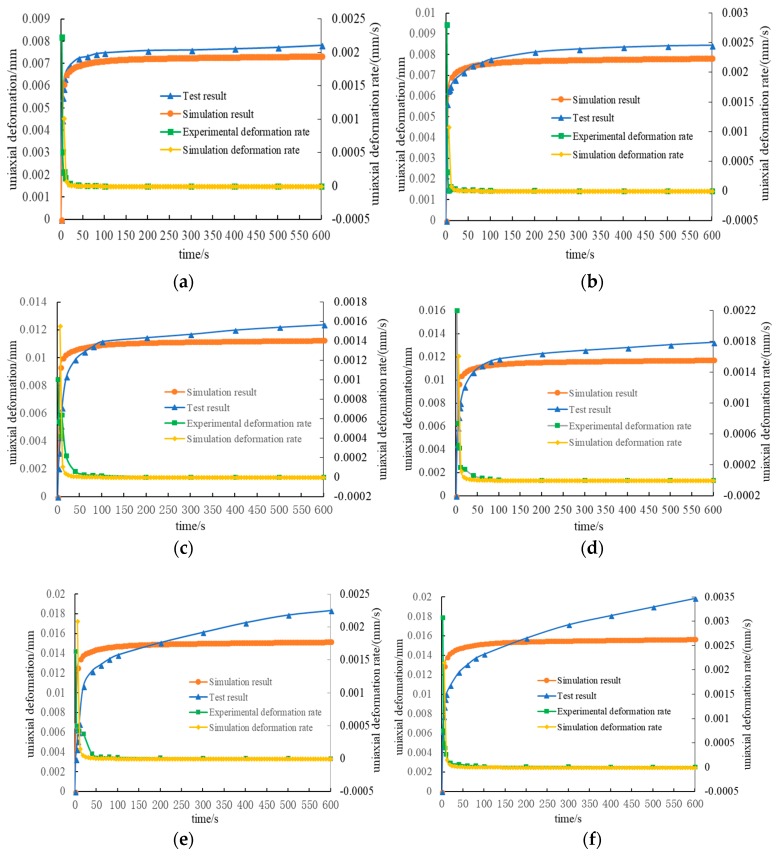
Experimental and simulation results of axial creep deformation under different loading conditions: (**a**) with 60 kPa confining pressure and 80 kPa axial loading; (**b**) without confining pressure and with 80 kPa axial loading; (**c**) with 60 kPa confining pressure and 120 kPa axial loading; (**d**) without confining pressure and with 120 kPa axial loading; (**e**) with 60 kPa confining pressure and 160 kPa axial loading; (**f**) without confining pressure and with 160 kPa axial loading.

**Figure 13 materials-12-02041-f013:**
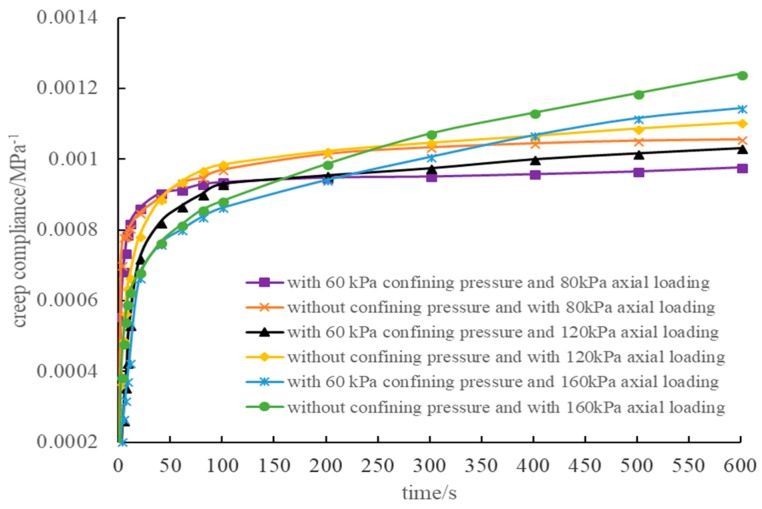
Experimental creep compliance curves under different loading conditions.

**Table 1 materials-12-02041-t001:** Percentage of materials used in asphalt mortar.

**Sieve Size (mm)**	**1.18–2.36**	**0.6–1.18**	**0.3–0.6**	**0.15–0.3**	**0.075–0.15**	**<0.075**	**Asphalt Binder**
**Percentage (%)**	30.3	15.5	15.0	7.6	2.6	15.8	13.2

**Table 2 materials-12-02041-t002:** Prony series coefficients of the relaxation modulus for asphalt mortar.

*i*	$\lambda_{n}$/s	$G_{n}$/MPa	$K_{n}$/MPa
1	10^−4^	1.1876	0.0212
2	10^−3^	8.2732	0.2124
3	10^−2^	7.1921	0.8938
4	10^−1^	39.3043	10.3226
5	10^0^	32.6392	20.2222
6	10^1^	28.9523	2.6625
7	10^2^	73.8451	29.5453
8	10^3^	307.0718	91.7736
		$G_{0}$ = 2.8219	$K_{0}$ = 0.0714

**Table 3 materials-12-02041-t003:** Loading scheme of triaxial creep test for asphalt mixture.

**Confining Pressure/kPa**	0			60		
**Axial Pressure/kPa**	80	120	160	80	120	160

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
