# Peer review of "Microstructural Modeling of Rheological Mechanical Response for Asphalt Mixture Using an Image-Based Finite Element Approach"

_materials, 2019, doi:10.3390/ma12132041_

Round 1
Reviewer 1 Report
The combination of experimental and numerical simulations is a very valuable approach. The subject matter submitted by the authors is important both in the research and application field. However, the content of the article can not find information on how many research samples the experimental part was carried out. What is the scattering of the measurements? I also suggest formatting the references in accordance with the requirements of the Journal, eg
Journal Articles:
1. Author 1, A.B .; Author 2, C.D. Title of the article. Abbreviated Journal Name Year, Volume, page range.
All tables and equations should be formatted in the same way.
Author Response
Reviewer #1 (Comments):
The combination of experimental and numerical simulations is a very valuable approach. The subject matter submitted by the authors is important both in the research and application field. However, the content of the article cannot find information on how many research samples the experimental part was carried out. What is the scattering of the measurements? I also suggest formatting the references in accordance with the requirements of the Journal, eg: Journal Articles: 1. Author 1, A.B .; Author 2, C.D. Title of the article. Abbreviated Journal Name Year, Volume, page range. All tables and equations should be formatted in the same way.
My specific comments are as follows:
1. How many research samples the experimental part was carried out?What is the scattering of the measurements?
Response: Thank you for your precious comments. In this paper, six tensile load levels and six torque shear stress levels were adopted to determine the tensile and shear stress thresholds of linear viscoelasticity for asphalt mortar. Three samples per load level were employed in determining Prony series coefficients of asphalt mortar viscoelastic constitutive model. The tensile and shear creep compliance curves of asphalt mortar are presented in Figure 1 and Figure 2. The differences of creep compliance for the three samples are small. In this paper, the average values of creep compliance for the three samples were employed in determining Prony series coefficients.
Fig. 1 Tensile creep compliance curves of asphalt mortar Fig. 2 Shear creep compliance curves of asphalt mortar
For triaxial creep experiment, two AC-20-type asphalt mixture specimens were adopted in the experiment, as can be seen in Figure 3. Only one specimen numerical and experimental results were compared in this study. However, the other specimen experimental results was better fitted with the numerical results within the linear viscoelastic scope. Due to the limited length of the article, only one specimen results was analyzed in this paper. We think that this comparison results presented in this paper should be sufficient to draw a conclusion that the three-dimensional microstructural model was able to effectively analyze the material mechanical response under triaxial load within the linear viscoelastic scope.
Fig. 3 AC-20-type asphalt mixture specimens
2. formatting the references in accordance with the requirements of the Journal.
Response: Thank you for your careful reading of our manuscript. The format of references have been revised according to the journal’s requirement.(On page 15-16, line 410-467)
3. All tables and equations should be formatted in the same way.
Response: We are sorry for this format error. We checked throughout the manuscript and made corrections where needed. (Eq. (1) to Eq. (19), Table 1-Table 3)
Again, special thanks to you for your good comments.

Reviewer 2 Report
(1) Abstract: (Line #19-25) The description on these sentences are not clear. The choice of words, i.e. classify, complicated, indirectly should be altered in a more specific expression.
(2) Introduction: The author(s) should mention clearly the novelty of this research among other previous researches.
(3) Abbreviations: The abbreviations should be spelled out one time: HMA, OTSU and so on.
(4) Use of '-' throughout the manuscript consistently.
(5) The caption of Figures are not specific and unclear such as 'Figure 1. The algorithm flow chart'.
(6) Notations in Eqs are not described all. For example, phi_t, phi_tau, and so on. What are the quantities of these? Is eij a deviatoric strain tensor? and J_o and B_o?
(7) Section 2.2.1 and 2.2.2: The constitutive model requires a graphical interpretation.
(8) Table 2: What is the lambda_n/S quantities? and what are the meaning of 10-4, 10-3,....?
(9) The informations on the loading plate are missing: rigid plate, frictional coefficients between the loading platen and speciemen.
(10) The readers are probably interested in the time step size rather than the step number in page 10.
(11) The monitoring points to measure the axial deformation should be addressed in page 10.
(12) Specify the source of experimental data used in Figure 10.
Author Response
My specific comments can be found in the uploaded PDF file.

Round 2
Reviewer 2 Report
All the corrections have been done accordingly.
Author Response
Again, special thanks to you for your good comments.